# Evolution of Estrogen Receptor Status from Primary Tumors to Metastasis and Serially Collected Circulating Tumor Cells

**DOI:** 10.3390/ijms21082885

**Published:** 2020-04-20

**Authors:** Carina Forsare, Pär-Ola Bendahl, Eric Moberg, Charlotte Levin Tykjær Jørgensen, Sara Jansson, Anna-Maria Larsson, Kristina Aaltonen, Lisa Rydén

**Affiliations:** 1Division of Oncology, Department of Clinical Sciences Lund, Lund University, SE-223 81 Lund, Sweden; carina.forsare@med.lu.se (C.F.); Par-Ola.Bendahl@med.lu.se (P.-O.B.); eric.moberg2@gmail.com (E.M.); charlotte.levin_tykjar_jorgensen@med.lu.se (C.L.T.J.); sara.jansson@med.lu.se (S.J.); anna-maria.larsson@med.lu.se (A.-M.L.); 2Division of Translational Cancer Research, Department of Laboratory Medicine, Lund University, SE-223 81 Lund, Sweden; kristina.aaltonen@med.lu.se; 3Division of Surgery, Department of Clinical Sciences Lund, Lund University, SE-223 81 Lund, Sweden; 4Department of Surgery, Skåne University Hospital, SE-222 42 Lund, Sweden

**Keywords:** circulating tumor cells, breast cancer, estrogen receptor: tumor progression, metastasis

## Abstract

Background: The estrogen receptor (ER) can change expression between primary tumor (PT) and distant metastasis (DM) in breast cancer. A tissue biopsy reflects a momentary state at one location, whereas circulating tumor cells (CTCs) reflect real-time tumor progression. We evaluated ER-status during tumor progression from PT to DM and CTCs, and related the ER-status of CTCs to prognosis. Methods: In a study of metastatic breast cancer, blood was collected at different timepoints. After CellSearch^®^ enrichment, CTCs were captured on DropMount slides and evaluated for ER expression at baseline (BL) and after 1 and 3 months of therapy. Comparison of the ER-status of PT, DM, and CTCs at different timepoints was performed using the McNemar test. The primary endpoint was progression-free survival (PFS). Results: Evidence of a shift from ER positivity to negativity between PT and DM was demonstrated (*p* = 0.019). We found strong evidence of similar shifts from PT to CTCs at different timepoints (*p* < 0.0001). ER-positive CTCs at 1 and 3 months were related to better prognosis. Conclusions: A shift in ER-status from PT to DM/CTCs was demonstrated. ER-positive CTCs during systemic therapy might reflect the retention of a favorable phenotype that still responds to therapy.

## 1. Introduction

Breast cancer is the most common malignant disease in women, and although five-year survival is approaching 90%, 20–30% of women with an initially local disease develop metastatic disease within 10 years [1]. The median overall survival (OS) in women with metastatic breast cancer (MBC) is approximately two years, and only 25% survive beyond five years [2]. Evaluating the response of therapy in MBC can be difficult, especially in patients with non-measurable disease. Guidelines for therapy evaluation include diagnostic imaging, clinical assessment, and also evaluation of liquid-based tumor markers [2,3,4].

In breast cancer, the estrogen receptor (ER) and the human epidermal growth factor receptor 2 (HER2), which are predictive of the response to endocrine- and HER2-targeted therapy, respectively, are the most important treatment predictive markers. Since it was shown that the expression of ER and HER2 frequently changes between primary tumor (PT) and distant metastasis (DM) in breast cancer [5,6], it is important to re-evaluate the biomarker status in biopsies from metastatic lesions. Furthermore, there are indications that cancer cells lose their ER expression during endocrine treatment [7,8] as a mechanism of resistance. The choice of therapy in a metastatic setting should therefore preferably be based on the receptor status of the metastatic biopsy when available. Due to heterogeneity, biopsies from different sites should be evaluated. In clinical practice, however, it is not always feasible to biopsy multiple metastatic sites, and the metastasis might also be situated at a non-accessible site. Moreover, current diagnostic methods, i.e., needle biopsies, limit the amount of material available for further analyses.

Liquid biopsies evaluating the presence of circulating tumor cells (CTCs) are non-invasive and easily accessible with a limited risk of complications in comparison to an invasive tissue biopsy. They can be repeated regularly, which allows for the serial monitoring of real-time tumor evolution/response to treatment. Additionally, a tissue biopsy reflects a momentary state of the tumor at one location in the body, whereas liquid-based markers reflect real-time tumor progression and are not dependent on repeated invasive tissue biopsies.

CTCs can be detected in a liquid biopsy, and the CellSearch^®^ system (^©^Menarini Silicon Biosystems, Inc 2020, USA) is considered the golden standard of CTC capturing and enumeration in MBC since it is the only Food and Drug Administration (FDA)-approved technology, so far. Multiple trials have shown CellSearch^®^ enumerated CTCs as an independent prognostic factor in MBC [9,10,11,12,13,14,15]. CTCs have also been proposed to more accurately reflect tumor heterogeneity than biopsies of single metastases and are considered to originate from both the primary tumor and all metastatic sites [16]. Therefore, liquid biopsies enumerating CTCs might be a better proxy for the total metastatic burden. However, CTC enumeration alone is insufficient to predict the efficacy of therapy, and the addition of CTC tumor-marker expression might better predict the benefit of a therapy [17].

The primary aim of the present study was to characterize CTCs detected in blood samples from MBC patients and to evaluate ER-status during tumor progression from primary tumors to distant metastasis and CTCs, respectively. Our secondary aim was to relate the ER-status of CTCs to prognosis. We hypothesized that, by evaluating the ER-status of single CTCs, improved therapy guidance could be provided.

## 2. Results

For the study cohort with an available CTC status using CellSearch^®^, and with information on the ER-status in the PT (*n* = 147), the median age was 65 years (range 40–84 years; Figure 1, Table 1). In total, 63% (92/147) of the patients were given endocrine therapy as first-line systemic therapy, whereas 48% (70/147) had chemotherapy. Furthermore, 84% (123/147) of PTs and 86% (105/122) of DMs were ER-positive (Table 1). The median time from diagnosis of PT to diagnosis of MBC was five years (range 0–36 years).

### 2.1. CTC Status at Baseline and During Follow-Up

At baseline (BL), 53% (76/144) of the patients were classified in the inferior prognostic group according to the previously approved and validated cut-off of ≥5 CTCs [9,10]. The corresponding percentages after 1 and 3 months of the first-line of therapy were 29% (37/128) and 19% (21/113), respectively.

DropMount slides could be prepared for 113 cases (Figure 1). When the CTC-DropMount method was used for CTC characterization, the corresponding numbers for cases included in the inferior prognostic group were 54% (35/65), 32% (7/22), and 52% (12/23) at BL, after 1 month and after 3 months of therapy, respectively.

### 2.2. ER Status of CTCs

By again applying the CTC-DropMount method for characterization, 26% (17/65) of CTCs were ER-positive at BL compared with 23% (5/22) after 1 month and 43% (10/23) after 3 months of therapy.

The distribution of number of CTCs at BL and the ER-status are depicted in Figure 2a, b. For ER-positive CTC cases at BL (Figure 2a), the number of detected cells ranged from 1 to 47, where cases were considered ER-positive if at least one CTC was positive for ER. With a cut-off of ≥5CTCs, 13 cases would have been regarded as ER-positive at BL instead of 17. For ER-negative cases (Figure 2b), the number of evaluated CTCs varied from 1 to 1094.

### 2.3. Shift in the ER Status

Evidence of a shift in the ER-status from PT to DM was assessed with the McNemar test, showing a shift towards more ER-negative tumors in DM (Table 2; Figure 3a). In total, 74% (90/122) retained ER positivity from PT to DM, 12% (15/122) changed from ER-positive in PT to ER-negative in DM, and 3% (4/122) changed from ER-negative to ER-positive DM (Table 2, Figure 3a).

Interestingly, when separately comparing the ER-status of the PT with the ER-status of CTCs at different timepoints, evidence of a shift from ER positivity to negativity was strong (Table 2; Figure 3b–d). Among 50 patients with ER-positive PTs, only 16 had ER-positive CTCs at BL (*p* < 0.0001; Figure 3b).

McNemar analysis comparing the ER-status in DM and of CTCs at different timepoints showed evidence of a negative association at BL and after 1 month but not after 3 months (Table 2).

Statistical power was low for a comparison of the ER-status of CTCs between follow-up timepoints (N_BL_ = 17, N_1 month_ = 19, and N_3 months_ = 10; Table 2).

### 2.4. ER Status of CTCs and PFS

Among all patients with positive CTCs (≥1 CTCs) at BL (*n* = 65), evidence of an association between the ER positivity of CTCs at BL, measured by the CTC-DropMount method, and improved PFS was weak (hazard ratio (HR): 0.57, 95% confidence interval (CI): 0.29–1.1, *p* = 0.10; Figure 4a). The effect, as well as the evidence, was stronger in landmark analyses using the ER-status of CTCs measured at 1 month (HR: 0.25, 95% CI: 0.055–1.1, *p* = 0.066; Figure 4b) and at 3 months (HR: 0.33, 95% CI: 0.11–1.01, *p* = 0.052; Figure 4c).

Similar results were seen in patients with ≥5 CTCs at BL (*n* = 61). Thirteen cases were defined as ER-positive and 48 cases as ER-negative CTCs at BL, and ER positivity was associated with better PFS at BL (HR: 0.33, 95% CI: 0.13–0.83, *p* = 0.019), at 1 month (HR: 0.64, 95% CI: 0.071–5.8, *p* = 0.69) and at 3 months (HR: 0.32, 95% CI: 0.078–1.3, *p* = 0.12).

## 3. Discussion

The tissue biopsy reflects a momentary state of a tumor at one location in the body, whereas liquid-based markers reflect real-time tumor progression and are not dependent on repeated invasive tissue biopsies. Enumeration and characterization of CTCs has potential as a less invasive diagnostic method to evaluate the progression of breast cancer. To our knowledge, this is one of a few studies evaluating the ER-status of single CTCs in metastatic breast cancer using an immunofluorescence method [18]. In the present study, we also evaluated the ER-status in PT and DM and identified a shift towards more ER-negative cases in DM. A shift in the ER-status from PT to CTCs was also detected. Out of 50 patients with ER-positive PT, only 16 had ER-positive CTCs at BL.

In the present study, retained ER positivity of CTCs after initiation of systemic therapy was associated with better prognosis and thus seemed to reflect a favorable retained phenotype that still responded to therapy. Van de Ven et al. showed that, after neoadjuvant chemotherapy, there was a 2.5–17% change in the ER-status from positive to negative [19]; in metastatic breast cancer, 17% of the cases were found to have a different ER-status in metastasis compared to the primary tumor [20]. Hence, we hypothesized that, by evaluating the ER-status of single CTCs, improved therapy guidance could be provided.

Bock et al. reported results similar to ours, with a shift towards more ER negativity in CTCs; 37% ER-positive in PT vs. 8% ER-positive in CTCs compared to our results, with 84% ER-positive in PT vs. 26% ER-positive in CTCs at BL [18]. Other studies showed ER-status discordance between CTCs and primary or metastatic biopsies similar to our results, ranging from 30% to 60% [21,22]. Lindström et al. showed an altered ER-status in one-third of patients during tumor progression from PT to DM, but CTCs were not assessed [23]. We identified a 16% (19/122) altered ER-status in PT vs. DM in the present study and four of them changed from ER-negative in PT to ER-positive in DM. Two of them had first-line endocrine therapy and had shorter time to progression than the median PFS in the study cohort. Unfortunately, the ER-status of the corresponding baseline CTCs was not available for these patients and consequently no conclusion on endocrine therapy resistance based on CTC data could be drawn. Furthermore, Lindström et al. stated that ER-positive recurrence had better prognosis than ER-negative recurrence, irrespective of the ER-status of the PT [23].

In our study, as treatment continued, the number of CTCs decreased, and this was confirmed by a decrease in the number of CTCs between different timepoints. However, we could not determine whether this decrease was due to technical issues with the CTC-DropMount method or a true biological difference. Fewer CTCs were found through immunofluorescence using the CTC-DropMount method than using the CellSearch^®^ method. This might have been because cells were attached to the walls of the CellSearch^®^ cartridge or were detached from the slides during the immunofluorescent-staining process. With fewer CTCs detected, the possibility of finding CTCs positive for ER consequently decreases. However, direct use of the CellSearch^®^ system for the evaluation of ER-status, as Paoletti et al. presented [24], would demand larger amounts of blood at a higher cost, which was not possible in this study. Other studies acknowledged technical difficulties concerning CTC characterization using fluorescence in situ hybridization [25]. With few cells to evaluate, this had the largest impact on the evaluation of ER-negative cases. The question was whether cases were truly negative or classified as such only because ER-positive cells were not detected.

According to the applied criteria for the definition of the ER-status of CTCs, only a few positive cells are needed for a sample to be considered ER-positive. Since CTCs disseminate from tumors, ER-negative cells could preferentially detach compared to ER-positive cells. This might explain the finding of a lower number of ER-positive CTCs than ER-positive PT. In line with our findings, Babayan et al. found 19% (9/16) ER-negative CTCs in patients who originally had ER-positive primary tumors [26].

As depicted in Figure 2, the number of detected cells ranged from 1 to 47 for samples classified as ER-positive. Regarding samples classified as ER-negative, on the other hand, the range was wider (1 to 1094). Ultimately, in cases where only one CTC was found, it is difficult to say whether this was or was not a true-negative CTC sample.

The strengths of our study include a prospective study design, with a homogeneous cohort (all patients were included before they started the first-line of systemic therapy), with predefined timepoints for sampling; consequently, the ER-status of CTCs was measured at longitudinal timepoints after the initiation of therapy. Furthermore, information on the ER-status of both PT and DM was available for most cases, enabling a thorough comparison throughout the tumor evolution. Weaknesses are mainly attributed to applying an in-house technique for capturing CTCs from the cartridge, which could potentially cause a lower cell recovery rate. Consequently, we had few cases to compare between the different timepoints, limiting the statistical power. Furthermore, there are few similar publications in the field with which to compare our results.

## 4. Material and Methods

The study was previously described in detail [27,28]. In brief, 168 women with newly diagnosed MBC, previously untreated for metastatic disease, were included in a prospective observational study at Skåne University Hospital and Halmstad County Hospital (ClinicalTrials.gov NCT01322893). Ethical permission was granted by the Lund University Ethics Committee (LU 2010/135) and all patients signed informed consent forms. Briefly, inclusion criteria were MBC diagnosis, age ≥18 years, Eastern Cooperative Oncology Group (ECOG) performance status score 0–2, and predicted life expectancy of >2 months. Study protocol included blood sampling at baseline (BL) and during therapy at 1, 3, and 6 months. In the current study, DropMount slides from BL (*n* = 113) and after 1 (*n* = 60) and 3 months (*n* = 46) were used. In total, 63% (92/147) of patients received adjuvant endocrine treatment, and 48% (70/147) received adjuvant chemotherapy. The primary endpoint was progression-free survival (PFS).

Patient and tumor characteristics (e.g., ER-status) were retrieved from clinical records and pathology reports. ER-status was routinely assessed by immunohistochemistry on formalin paraffin-embedded material (with commercially available antibodies) on surgical specimen from the primary tumors at the time of primary diagnosis and on diagnostic core biopsies from metastasis at the time of diagnosis of metastatic disease. All assessments were performed by board-certified pathologists and ≥10% were classified as ER-positive according to the European guidelines [2,29].

After CTC enumeration using CellSearch^®^, cells were fixated using the CTC-DropMount method as previously described [30]. In brief, isolated CTCs were separated using a magnetic stand, and all nonmagnetic fluid was removed. The remaining cells were resuspended in phosphate-buffered saline (PBS), dropped onto Superfrost™ slides (ThermoScientific™, Germany), dried at 37 °C for 30 min, and fixated in pure methanol for 5 min. The slides were stored at –20 °C until analysis (i.e., CTC-DropMount method).

With at least five CTCs enumerated using CellSearch^®^, 113 cases were available as DropMount at BL. Using a previously described immunofluorescence method [30], cells were evaluated in the present study for ER expression at BL and after 1 and 3 months. A rabbit monoclonal anti-ERα antibody was used as a primary (Thermo Fisher Scientific, #9101S1210D) and an AlexaFluor488-labeled goat antirabbit as a secondary antibody (Life Technologies, #1423009). For CD45 staining, a rat monoclonal antibody was used as a primary (Thermo Fisher Scientific/Invitrogen, #MA5-17687) and an AlexaFluor647-labeled goat antirat as a secondary antibody (Thermo Fisher Scientific/Invitrogen, #A-21247). Cytokeratins 8, 9, and 19 were stained with phycoerythrin-labeled mouse monoclonal antibody provided as leftovers from the CellSearch^®^-kit (Menarini, #E491A). Blood spiked with cells from the MCF-7 cell line was used as a positive control for ER expression. Cells positive for 4’,6-diamidino-2-phenylindole (DAPI) and cytokeratins 8, 18, and 19 but negative for CD45 were considered to be CTCs. Samples were classified as ER-positive if at least one CTC was positive for ER. Assessment of the ER-status of CTCs was independently performed by EM and CF. The cells were scanned using a BX63 Upright Microscope (Olympus Corporation, LRI, Lund, Sweden) and CellSense Dimension software. The filters used were DAPI, Cy2/fluorescein isothiocyanate (FITC), Cy3/tetramethylrhodamine-5-isothiocyante (TRITC), and Cy5. Scans were performed at 20×, while objects of special interest were captured at 40×. A representative example is shown in Figure 5.

### Statistical Analysis

The study followed the Reporting Recommendation for Tumor Marker (REMARK) criteria [31,32]. Kaplan–Meier plots and the log-rank test were used to compare PFS between patient groups. Survival analysis of ER expression measured at 1 and 3 months was performed with landmark analysis. Hazard ratios (HRs) were quantified by the univariable Cox proportional hazards-regression model, and proportional hazard assumptions were graphically checked. Comparison of the ER-status of PT, DM, and CTCs at different timepoints was performed using the McNemar test. Stata version 16.1 (StataCorp, College Station, TX, USA) was used for statistical analyses.

## 5. Conclusions

We demonstrated evidence of a shift from ER positivity to negativity in the PT compared to DM, as well as evidence of a similar shift in the ER-status of CTCs from PT and DM. Using the CTC-DropMount method, the ER positivity of CTCs at 1 and 3 months was related to better prognosis. The ER positivity of CTCs after the initiation of systemic therapy might reflect the retention of a favorable phenotype that still responds to therapy.

## Figures and Tables

**Figure 1 ijms-21-02885-f001:**
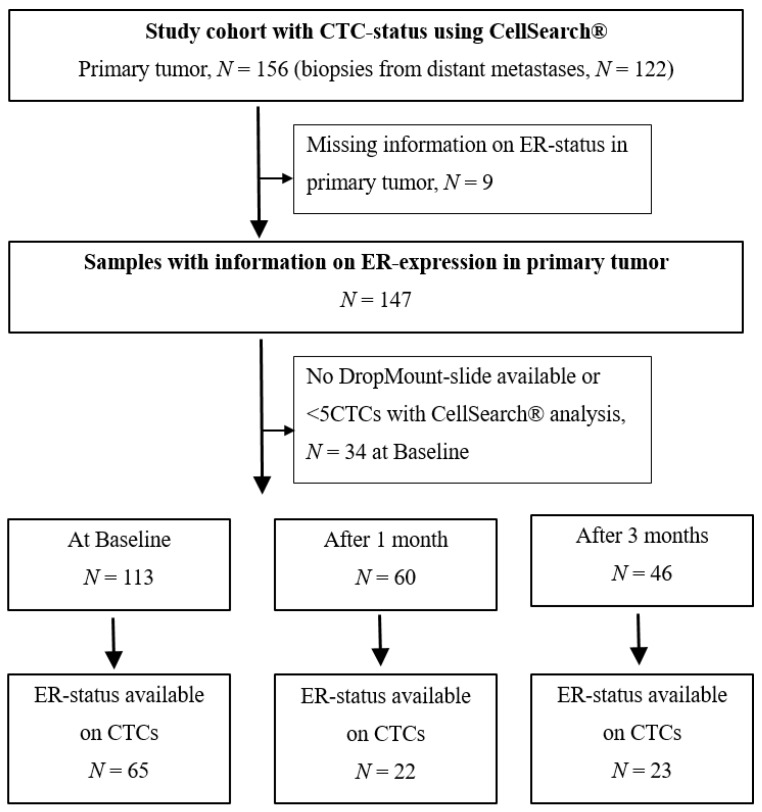
Flowchart of study cohort. CTC = circulating tumor cell, ER = estrogen receptor.

**Figure 2 ijms-21-02885-f002:**
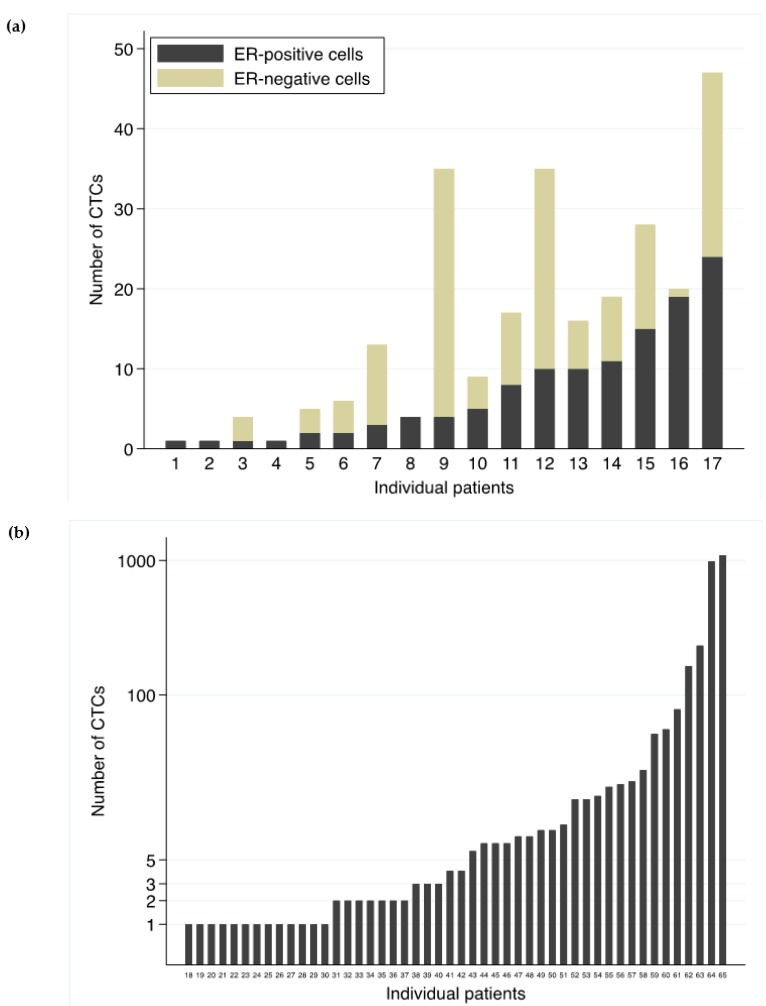
ER-status of CTCs. Depicted are number of evaluated circulating tumor cells (CTCs) for individual cases at baseline (BL) and the corresponding estrogen receptor (ER) distribution among the evaluated CTCs. Seventeen cases (1–17) were defined as ER-positive (**a**) and 48 cases (18–65) as ER-negative. Note that the scale in (**b**) is logarithmic due to better visuality.

**Figure 3 ijms-21-02885-f003:**
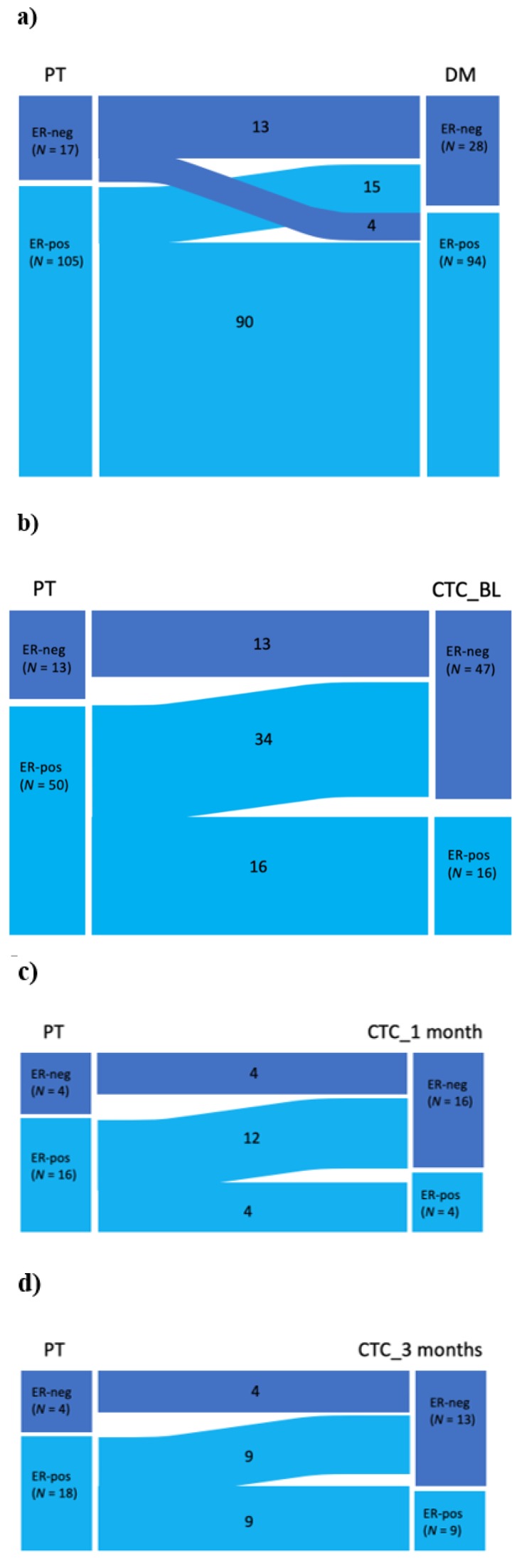
Shift in the ER-status. Sankey diagrams depicting shifts in the estrogen receptor (ER) status from (**a**) primary tumor (PT) to distant metastasis (DM), (**b**) from PT to circulating tumor cells (CTCs) at baseline (BL), (**c**) from PT to CTCs at 1 month, and (**d**) from PT to CTCs at 3 months.

**Figure 4 ijms-21-02885-f004:**
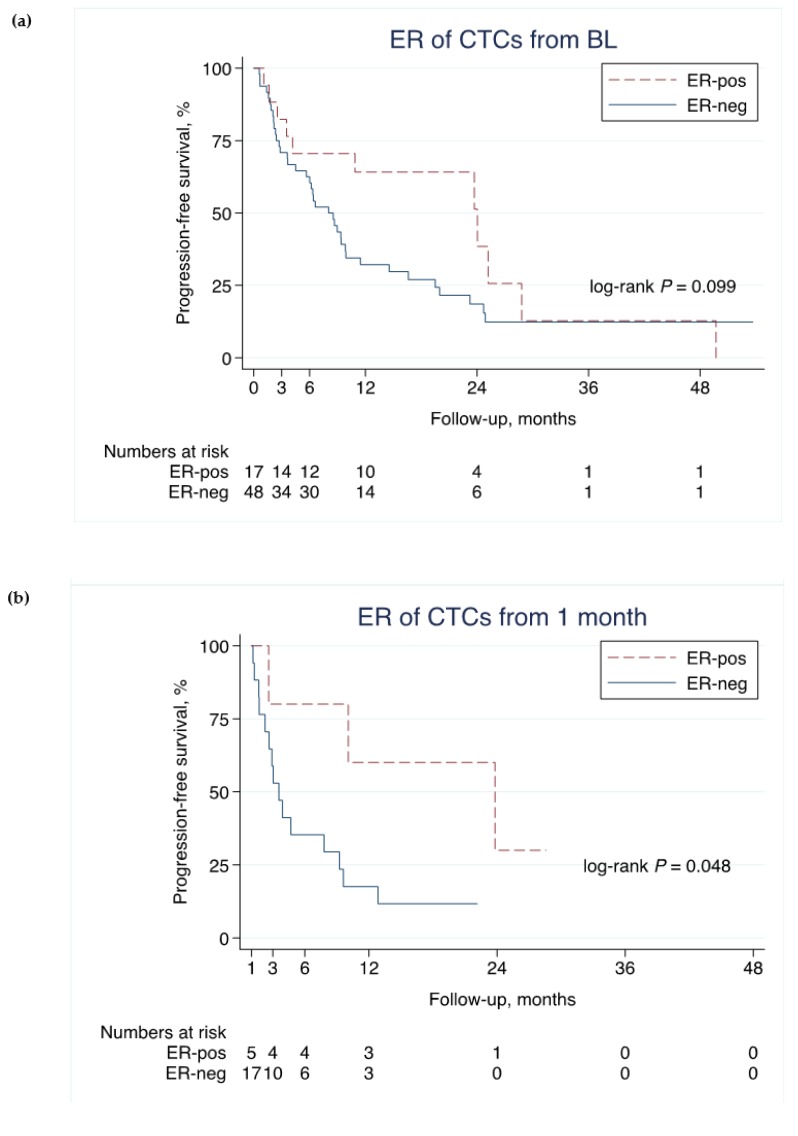
ER-status on CTCs and PFS. Association of progression-free survival (PFS) and the estrogen receptor (ER) status at different timepoints using the circulating tumor cell (CTC) Drop-Mount method. Note that the time-axes start at 0, 1, and 3 months, respectively, in (**a**), (**b**), and (**c**) (landmark analysis). * *p*-values are from the log-rank test.

**Figure 5 ijms-21-02885-f005:**
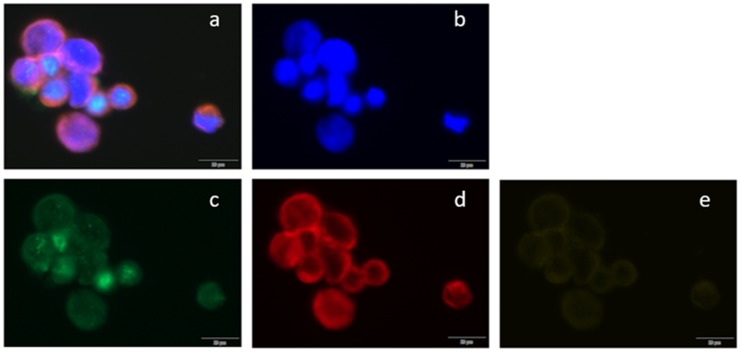
Immunofluorescent staining of CTCs in metastatic breast cancer blood samples. Circulating tumor cells (CTCs) were scanned using (**a**) BX63 Upright Microscope; (**a**) a composite image of all channels, (**b**) 4’,6-diamidino-2-phenylindole (DAPI) counterstain (fluorescent blue), (**c**) estrogen receptor α stained with AlexaFluor488 (green), (**d**) cytokeratins 8, 18, and 19 stained with phycoerythrin (red), and (**e**) CD45 stained with AlexaFluor647 (yellow).

**Table 1 ijms-21-02885-t001:** Patient and tumor characteristics for *n* = 147 with an available estrogen receptor (ER) status in the primary tumor (PT). CTC = circulating tumor cell, BL = baseline, DM = distant metastases, MBC = metastatic breast cancer, HER2 = human epidermal growth factor receptor 2, NHG = Nottingham histological grade.

	All Patients, PT (*n* = 147)	CTC BL (*n* = 113)	CTC 1 Month (*n* = 60)	CTC 3 Months (*n* = 46)	DM after PT (*n* = 122)
**Age at diagnosis of MBC**					
Median (range) in years	65 (40–84)	60 (34–83)	59 (37–82)	59 (39–77)	59 (32–83)
**Time from PT to DM**					
Median (range) in years	4.5 (0–36)	4.2 (0–36)	3.6 (0–36)	4.7 (0–36)	5.2 (0–36)
**PT ER-status, *n* (%)**					
Negative	24 (16%)	17 (15%)	8 (13%)	6 (13%)	17 (14%)
Positive	123 (84%)	93 (85%)	52 (87%)	40 (87%)	105 (86%)
Missing		3		1	
**DM ER-status, *n* (%)**					
Negative	28 (23%)	22 (23%)	13 (25%)	7 (18%)	28 (23%)
Positive	94 (77%)	73 (77%)	39 (75%)	31 (82%)	94 (77%)
Missing	25	18	8	8	25
**HER2-status, *n* (%)**					
Negative	109 (87%)	83 (87%)	50 (94%)	39 (97%)	88 (86%)
Positive	17 (13%)	11 (13%)	3 (6%)	1 (3%)	14 (14%)
Missing	21	19	7	6	20
**PT tumor size, *n* (%)**					
T1	52 (37%)	35 (34%)	16 (27%)	15 (35%)	50 (42%)
T2	49 (35%)	37 (36%)	23 (40%)	14 (33%)	43 (36%)
T3	20 (14%)	16 (16%)	8 (14%)	7 (16%)	15 (13%)
T4	19 (14%)	15 (14%)	11 (19%)	7 (16%)	10 (9%)
Missing	7	10	2	3	4
**PT Node status, *n* (%)**					
Node negative	41 (32%)	27 (28%)	13 (25%)	10 (24%)	39 (35%)
Node positive	88 (68%)	72 (72%)	39 (75%)	30 (76%)	74 (65%)
Missing	18	14	8	6	9
**PT NHG, *n* (%)**					
Grade 1	12 (10%)	8 (9%)	2 (4%)	4 (10%)	11 (10%)
Grade 2	64 (53%)	52 (58%)	31 (63%)	24 (62%)	60 (56%)
Grade 3	45 (37%)	29 (33%)	16 (33%)	11 (28%)	37 (34%)
Missing	26	24	11	7	14
**First-line systemic treatment, *n* (%)**					
Endocrine only	92 (63%)	67 (59%)	35 (58%)	33 (72%)	86 (70%)
Chemotherapy	70 (48%)	53 (47%)	33 (53%)	32 (70%)	63 (52%)
HER2-targeted	5 (3%)	1 (1%)	1 (2%)	1 (2%)	5 (4%)
**CellSearch^®^ BL CTC number**					
Median (range)	5 (0–2598)	17 (0–2598)	42 (0–2598)	42 (0–2598)	5 (0–2598)
Mean	678	88	118	119	74

**Table 2 ijms-21-02885-t002:** McNemar analysis comparing the ER-status in the primary tumor (PT) vs. the estrogen receptor (ER) status of distant metastases (DM) and circulating tumor cells (CTCs) at different timepoints; baseline (BL), 1 month and 3 months.

ER Status	PT vs. DM	PT vs. BL	PT vs. 1 Month	PT vs. 3 Months	DM vs. BL	DM vs. 1 Month	DM vs. 3 Months	BL vs. 1 Month	BL vs. 3 Months	1 Month vs. 3 Months
	*n* = 122 (%)	*n* = 63 (%)	*n* = 20 (%)	*n* = 22 (%)	*n* = 55 (%)	*n* = 17 (%)	*n* = 20 (%)	*n* = 17 (%)	*n* = 19 (%)	*n* = 10 (%)
-/-	13 (11)	13 (21)	4 (20)	4 (18)	12 (22)	5 (29)	6 (30)	10 (59)	7 (37)	5 (50)
-/+	4 (3)	0	0	0 (0)	0	0	1 (5)	3 (17)	6 (31)	1 (10)
+/-	15 (12)	34 (54)	12 (60)	9 (41)	29 (53)	7 (41)	4 (20)	2 (12)	3 (16)	2 (20)
+/+	90 (74)	16 (25)	4 (20)	9 (41)	14 (25)	5 (29)	9 (45)	2 (12)	3 (16)	2 (20)
*p-value*	*0.019*	*<0.0001*	*0.0005*	*0.004*	*<0.0001*	*0.016*	*0.38*	*1.0*	*0.51*	*1.0*

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
