# Peer review of "Evolution of Estrogen Receptor Status from Primary Tumors to Metastasis and Serially Collected Circulating Tumor Cells"

_ijms, 2020, doi:10.3390/ijms21082885_

Round 1
Reviewer 1 Report
many compliments for your paper which demonstrates clarity and originality, even if the sample size is not so big. It's worthy of being published after a minor revision:
line 16-17: can you introduce a reference?
in figure 3 there is an error. 34 ER + patients must converge into ER -
Author Response
- line 16-17: can you introduce a reference?
Answer: Although no page is provided, we suggest the missing reference is on page 2 and we have added a reference to the published SWOG S0500-study
Smerage JB, Barlow WE, Hortobagyi GN, Winer EP, Leyland-Jones B, Srkalovic G, Tejwani S, Schott AF, O'Rourke MA, Lew DL, et al. Circulating tumor cells and response to chemotherapy in metastatic breast cancer: SWOG S0500. J Clin Oncol. 2014;32(31):3483–9.
- in figure 3 there is an error. 34 ER + patients must converge into ER –
Answer: Thank you for pointing this out, we have now corrected figure 3b.
Reviewer 2 Report
This is an interesting manuscript describing a prospective clinical study designed (among other things) to evaluate estrogen receptor (ER) status during breast cancer progression from primary tumors to distant metastasis and circulating tumor cells (CTCs). Although the novelty of the study is fairly modest, the trial design is strong and the data are robust and well-presented. Strengths and weaknesses of the study are also thoughtfully considered, which is appreciated. Overall, the manuscript therefore represents a useful addition to the existing literature. Attention to a few items would improve the manuscript, and these are described below.
- Overall, the manuscript would benefit from English language/grammar review and editing.
- Lines 35-36: “….which are predictive of endocrine- and HER2-targeted therapy, respectively” should be changed to “….which are predictive of response to endocrine- and HER2-targeted therapy, respectively”.
- Line 42: “…In clinical praxis” should be changed to “In clinical practice”.
- In Figure 2, it is not clear what the numbers on the X-axis represent. Please include an axis label and clarify this in the figure legend.
- More detail should be provided in the Materials & Methods with regards to acquisition of and analysis of biopsy tissues from distant metastases.
- Of the 3% (4/122) of patients that changed from ER-negative in the primary tumor to ER-positive in distant metastasis, were anti-estrogen therapies given to these patients, and if so, was there a clinical response? Even if treatment was not changed in these patients, the Discussion would benefit from consideration about how ER-status in CTCs might be incorporated into clinical practice in order to help direct therapy, particularly for patients who develop resistance to their current therapy.
Author Response
- Overall, the manuscript would benefit from English language/grammar review and editing.
Answer: The manuscript has been through a language editing service recommended by MDPI – please find attached the invoice for the English editing (Specialist edit) service, By changing all phrasing pointed out by the reviewers, we hope the language has been further improved.
- Lines 35-36: “….which are predictive of endocrine- and HER2-targeted therapy, respectively” should be changed to “….which are predictive of response to endocrine- and HER2-targeted therapy, respectively”.
Answer: Thank you for pointing this out, we have corrected the phrasing according to the comment.
- Line 42: “…In clinical praxis” should be changed to “In clinical practice”.
Answer: This has also been changed according to the recommendation.
- In Figure 2, it is not clear what the numbers on the X-axis represent. Please include an axis label and clarify this in the figure legend.
Answer: Thank you for pointing out this to us. The x-axis represent individual cases 1-17 in a) and 18-65 in b). This has now been clarified in the legend and figure text.
- More detail should be provided in the Materials & Methods with regards to acquisition of and analysis of biopsy tissues from distant metastases.
Answer: We have now clarified this by rephrasing the sentence.
ER-status was routinely assessed by immunohistochemistry on formalin paraffin-embedded material (with commercially available antibodies) on surgical specimen from primary tumors at time of primary diagnosis and on diagnostic core biopsies from metastasis at time of diagnosis of metastatic disease. All assessments were performed by board-certified pathologists and 10% were classified as ER-positive according to the European guidelines [2, 28].
- Of the 3% (4/122) of patients that changed from ER-negative in the primary tumor to ER-positive in distant metastasis, were anti-estrogen therapies given to these patients, and if so, was there a clinical response? Even if treatment was not changed in these patients, the Discussion would benefit from consideration about how ER-status in CTCs might be incorporated into clinical practice in order to help direct therapy, particularly for patients who develop resistance to their current therapy.
Answer: Thank you for this interesting comment. We have checked the initial systemic therapy for these patients and 2 out of 4 were given endocrine treatment. The two patients who started on endocrine therapy as first-line systemic therapy had shorter progression-free survival, 2.8 months (tamoxifen) and 4.5 months (aromatase inhibitor) than median PFS of the study cohort (N = 122) which was 10.7 months. ER-status in corresponding baseline CTCs were not available for these patients and thus no conclusion on endocrine therapy resistance based on CTC data can be drawn.
We have added a note on page 9, line 25:
We identified 16% (19/122) altered ER-status in PT vs. DM in the present study and four of them changed from ER-negative in PT to ER-positive in DM. Two of them had first-line endocrine therapy and had shorter time to progression than median PFS in the study cohort. Unfortunately, ER status in corresponding baseline CTCs was not available for these patients and thus no conclusion on endocrine therapy resistance based on CTC data can be drawn.